# Biocontrol of Cercospora leaf spot in sugar beet by a novel *Bacillus velezensis* KT27 strain: Enhanced antifungal activity and growth promotion in laboratory and field conditions

Agnieszka Wita[1], Wojciech Białas[1], Katarzyna Czaczyk[1], Agnieszka Drożdżyńska[1], Łukasz Sobiech[2], Monika Grzanka[2], Jakub Danielewicz[3], Ewa Jajor[3], Joanna Horoszkiewicz[3], Roman Marecik [1]*

1 Department of Biotechnology and Food Microbiology, Poznań University of Life Sciences, Poznań, Poland, 2 Department of Agronomy, Poznań University of Life Sciences, Poznań, Poland, 3 Department of Mycology, Institute of Plant Protection, National Research Institute, Poznań, Poland

* roman.marecik@up.poznan.pl

## Abstract

Diseases in crops are a major contributor to yield reduction and economic losses. Cercospora leaf spot (CLS), caused by *Cercospora beticola*, is among the most severe diseases affecting sugar beet and other crops. The increasing resistance of *C. beticola* to conventional chemical fungicides, along with their excessive application, exacerbates environmental pollution. This study investigates the antagonistic activity of a newly isolated strain, *Bacillus velezensis* KT27, against *Cercospora beticola*, *Rhizoctonia cerealis*, and *Fusarium oxysporum* under laboratory conditions. The bacterium's ability to produce lipopeptides (surfactin, iturin, and fengycin) and solubilize phosphorus, potassium, and zinc was also assessed. In vitro assays revealed that *B. velezensis* KT27 effectively inhibited *C. beticola* growth (60.2%), though it exhibited lower antagonistic activity against *R. cerealis* (22.5%) and *F. oxysporum* (15.5%). The elimination of bacterial biomass by centrifugation and the use of sterile supernatant reduced antifungal activity by more than 3.5-fold for all tested fungi, highlighting the importance of direct bacterial interactions. Notably, the antagonistic effect of *B. velezensis* KT27 against *C. beticola* significantly increased when bacterial cultures were supplemented with thermally inactivated fungal biomass of *C. beticola* especially *R. cerealis*. Field experiments demonstrated the high efficacy of *B. velezensis* KT27 biological control agent, particularly when induced by *R. cerealis*. The level of CLS protection achieved with the bacterial treatment was only 9.1% lower than that obtained using a combination of three chemical fungicides. Additionally, the biocontrol agent positively influenced sugar beet growth, leading to a root yield increase of up to 15.2% compared to the untreated control. These findings highlight the potential of *B. velezensis* KT27 as an effective and environmentally sustainable biocontrol agent against CLS in sugar beet cultivation.

**Data availability statement:** All relevant data underlying the paper are deposited in the public repository ZENODO, and it is available at DOI: 10.5281/zenodo.15338237.

**Funding:** The author(s) received no specific funding for this work.

**Competing interests:** The authors have declared that no competing interests exist.

## Introduction

Over recent decades, agricultural intensification has played a pivotal role in ensuring global food security in response to a rapidly growing population. This intensification has predominantly relied on successive generations of chemical agents to manage weeds, pests, and plant diseases [1]. However, the long-term and widespread use of synthetic agrochemicals has contributed significantly to environmental degradation, including contamination of soil, air, and groundwater, as well as eutrophication of aquatic ecosystems. Moreover, chemical residues in plant tissues pose direct health risks to both humans and animals [2,3].

In response to these challenges, the European Commission introduced the European Green Deal, which advocates for sustainable agricultural practices that prioritize soil health, biodiversity restoration, and reduced dependence on mineral fertilizers and chemical plant protection products [4,5]. This policy direction has revived interest in research initiated in the mid-20th century, highlighting the beneficial effects of specific microorganisms on plant and animal productivity [5,6]. A global trend has emerged, favoring the integration of biological alternatives or supplements to chemical plant protection, as evidenced by the growing market value of microbial bioactive agents [7]. Consequently, there is increasing demand for innovative, eco-friendly solutions to replace conventional agrochemical practices with approaches that maintain environmental integrity and promote agricultural sustainability [8]. In this context, biological methods of plant protection and productivity enhancement have garnered substantial attention. These strategies leverage the diverse interactions among microorganisms to suppress phytopathogens via mechanisms such as antagonism, nutrient competition, parasitism, and antibiosis [9]. Certain microbial strains, employed as biocontrol agents, produce bioactive compounds including volatile organic compounds (VOCs), antibiotics, lipopeptides, hydrolytic enzymes, and phytohormones, all of which contribute to improved crop performance [10–12]. Furthermore, many of these microbes exhibit plant growth-promoting traits, such as nitrogen fixation, mineral solubilization (e.g., phosphorus, potassium, silica), and siderophore production, enhancing nutrient availability [13].

Among these beneficial microorganisms, Plant Growth-Promoting Bacteria (PGPB), particularly those belonging to the genus *Bacillus*, have shown significant potential for sustainable crop management [14,15]. *Bacillus* is a diverse genus of Gram-positive, spore-forming bacteria, commonly isolated from soil, freshwater, air, and plant or animal matter. Their exceptional adaptability allows survival under diverse conditions, including extremes of temperature, salinity, oxygen availability, and pH. The genus is also characterized by a broad metabolic repertoire, exploited industrially for enzyme production, pharmaceutical synthesis, and biotechnological applications [16]. While some *Bacillus* species, such as *B. cereus* and *B. anthracis*, are pathogenic to humans, others—such as *B. subtilis*, *B. amyloliquefaciens*, *B. pumilus*, and *B. thuringiensis*—have been successfully utilized in agriculture as biocontrol agents [17,18]. These strains suppress fungal, bacterial, and nematode pathogens and form highly resilient endospores that confer long-term environmental persistence, making them especially suitable for field applications [19]. One *Bacillus*

species of particular interest is *Bacillus velezensis*, first isolated from environmental samples at the mouth of the Vélez River in Torredelmar, Malaga, Spain (strain CR-502T and CR-14b) [20]. Subsequent studies have identified this species in diverse ecological niches, including fish farm wastewater and poultry gastrointestinal tracts [12,21]. Phylogenetic analyses reveal its close genetic affinity to *B. subtilis* and *B. amyloliquefaciens* [21]. Whole-genome analyses of various *B. velezensis* strains have unveiled their capacity to produce a broad spectrum of secondary metabolites with antimicrobial properties and tolerance to abiotic stress [22–24]. Notably, strain HNA3 synthesizes 14 distinct secondary metabolites, including lipopeptides (fengycin, bacillomycin D, surfactin, mycosubtilin, bacillibactin), polyketides (macrolactin, difficidin, bacillaene), antibacterial peptides (bacilysin, amylocyclin), and others such as plipastatin, iturin, and paenilarvins [23,24]. These strains also produce phytohormones like indole-3-acetic acid (IAA), which promotes plant growth [23], and volatile compounds such as 2,3-butanediol, further contributing to plant health [25]. In addition, strains like SQR9 colonize plant roots and secrete metabolites that stimulate native microbial communities, facilitating biofilm formation [26].

Due to these multifaceted traits, *B. velezensis* has become a prominent biological alternative to conventional fertilizers and fungicides. Its formulations have demonstrated efficacy against multiple plant diseases, including wheat powdery mildew, Fusarium head blight (FHB), and Phytophthora-induced pathologies [18]. Current research efforts focus on identifying new *B. velezensis* strains with enhanced biocontrol efficacy, optimizing their formulations for field application, and exploring synergistic interactions with other beneficial organisms, including fungi, to further improve antifungal activity [27,28].

Among the most devastating fungal diseases affecting sugar beet (*Beta vulgaris* ssp. *vulgaris*) is Cercospora leaf spot (CLS), caused by *Cercospora beticola*. This pathogen also affects other species within the *Beta* genus and members of the *Chenopodiaceae, Acanthaceae, Apiaceae, Asteraceae, Brassicaceae, Malvaceae, Plumbaginaceae*, and *Polygonaceae* families [29]. The disease initially targets older foliage and progresses to younger leaves, manifesting as circular lesions that vary in color from light brown and grey to reddish-purple or yellow. These necrotic spots reduce the photosynthetic capacity of the plant, leading to wilting and, eventually, leaf death. Yield losses can reach up to 40%, and infected plants produce lower-quality raw material for sugar extraction [30]. The persistence of pathogen structures such as pseudostromata in the soil further complicates management, as they remain viable for up to two years [31,32]. Although crop rotation and the use of resistant cultivars are recommended, chemical fungicides remain the most widely used control measure. However, resistance of *C. beticola* to commonly used active ingredients has been increasingly documented, necessitating higher dosages or alternative compounds, thus exacerbating environmental and economic concerns [33,34].

This study investigates the newly isolated *B. velezensis* KT27 strain and its potential application as a biocontrol agent against *C. beticola*, the causal agent of CLS in sugar beet. The antagonistic activity of *B. velezensis* KT27 was assessed both with and without prior induction using inactivated cultures of *C. beticola, Rhizoctonia cerealis,* and *Fusarium oxysporum*. Additionally, its inhibitory effects against *R. cerealis* and *F. oxysporum* were also evaluated. The findings offer promising insights into the development of environmentally sustainable plant protection strategies using microbial biocontrol agents.

## Materials and methods

### Microorganism strains

The bacterial strain *Bacillus velezensis* KT27, used in this study, was isolated from compost derived from the fermentation and composting of selectively collected biodegradable waste. The compost material included green waste from the maintenance of public green areas, gardens, and parks, as well as kitchen waste. Bacterial isolation was carried out using the standard serial dilution technique. Briefly, 1 g of compost was suspended in 10 mL of sterile saline solution and thoroughly mixed. Serial ten-fold dilutions were prepared, and 1 mL aliquots from each dilution were spread onto Petri dishes containing Tryptone Soya Agar (TSA; Becton, Dickinson and Co., Franklin Lakes, NJ, USA). The plates were incubated at 30°C for 48 hours. Distinct colonies were subsequently selected and subcultured on fresh TSA plates to obtain pure isolates. Purified strains were stored in glycerol stocks at –80°C for long-term preservation, following the method described by [35].

Initial characterization of the *B. velezensis* KT27 strain was based on colony morphology, followed by identification using matrix-assisted laser desorption/ionization time-of-flight mass spectrometry (MALDI-TOF MS). Molecular confirmation was performed through partial sequencing of the 16S rRNA gene, according to the procedure described by Wita et al. [36]. Genomic DNA was extracted from overnight bacterial cultures using the Genome Mini AX Bacteria Kit (A&A Biotechnology, Gdańsk, Poland), following enzymatic pretreatment with lysozyme (50 mg/mL; Sigma-Aldrich) for 1 hour at 37°C. PCR amplification of the 16S rRNA gene fragment was conducted, and PCR products were purified using the Clean-up Kit (A&A Biotechnology). Sequencing was outsourced to Genomed (Warsaw, Poland). The resulting sequences were assembled into contigs and analyzed using the BLAST+ 2.15.0 algorithm against the NCBI GenBank database for taxonomic identification.

Phytopathogenic fungal strains, including *Cercospora beticola*, *Fusarium oxysporum*, and *Rhizoctonia cerealis*, were kindly provided by the Institute of Plant Protection – National Research Institute (Poznań, Poland).

## Cultivation of microorganisms

The *B. velezensis* KT27 strain was cultured in Tryptone Soya Broth (TSB; Becton, Dickinson and Co., Franklin Lakes, NJ, USA) for 48 hours at 30°C with continuous agitation on a rotary shaker set at 250 rpm. Bacterial cell concentration was determined using the standard pour plate method. Aliquots were plated onto Tryptone Soya Agar (TSA; Becton, Dickinson and Co.) and incubated at 30°C for 48 hours. Colonies were counted manually, and bacterial abundance was expressed as colony forming units per milliliter (CFU/mL).

Phytopathogenic fungi, including *F. oxysporum*, *R. cerealis*, and *C. beticola*, were cultivated in Yeast Extract Peptone Dextrose (YPD) broth for 5 days at 26°C on a rotary shaker set at 180 rpm, following the protocol described by Nandhini, et al [37].

## Antifungal efficacy of *B. velezensis* KT27

The antifungal activity of *B. velezensis* KT27 was evaluated against *C. beticola*, *F. oxysporum*, and *R. cerealis*, which are recognized as major fungal pathogens responsible for significant crop yield losses. Following 48 hours of bacterial cultivation, the culture reached a concentration of approximately $1.0 \times 10^8$ CFU/mL. The bacterial suspension was then divided into two portions: one was used directly, while the other was centrifuged at 4,500 rpm for 10 minutes. The resulting supernatant was filtered through a 0.45 µm pore-size syringe filter (Merck, Darmstadt, Germany) to obtain a sterile cell-free filtrate. Antagonistic activity was assessed using the agar well diffusion method described by Balouiri et al. [38]. Petri dishes were prepared with 20 mL of molten Potato Dextrose Agar (PDA; Oxoid, Milan, Italy) cooled to 55°C and supplemented with 2% (v/v) fungal inoculum. After solidification, two wells (10 mm diameter) were punched into the agar. One well was filled with 100 µL of the bacterial culture, and the other with 100 µL of the sterile supernatant. A negative control well was filled with 100 µL of sterile deionized water. Plates were incubated at 26°C for four days. Post incubation, inhibition zones surrounding each well were measured in millimeters. The percentage of fungal growth inhibition was calculated relative to the control. Representative results demonstrating the antagonistic activity of *B. velezensis* KT27 are shown in Fig 1.

## Enhancement of *B. velezensis* KT27 activity against *C. beticola*

Based on results from a separate, unpublished study, the growth medium for *B. velezensis* KT27 was modified to enhance its antagonistic activity. Tryptone Soya Broth (TSB) was supplemented with 20 g/L of mannitol and 5 g/L of yeast extract. To further stimulate bacterial metabolic activity, the medium was enriched with 10% (v/v) of heat-inactivated fungal cultures (*C. beticola*, *F. oxysporum*, or *R. cerealis*), previously autoclaved at 121°C for 30 min. The modified medium was inoculated with *B. velezensis* KT27 at a concentration of $1.0 \times 10^8$ CFU/mL, corresponding to 5% of the total culture volume. Cultivation was performed at 30°C for 48 hours under constant agitation (250 rpm) on a rotary shaker. The resulting cultures were subsequently assessed for their antagonistic activity against *C. beticola* using the well diffusion method as described by Balouiri et al. and Kim et al. [35,38].

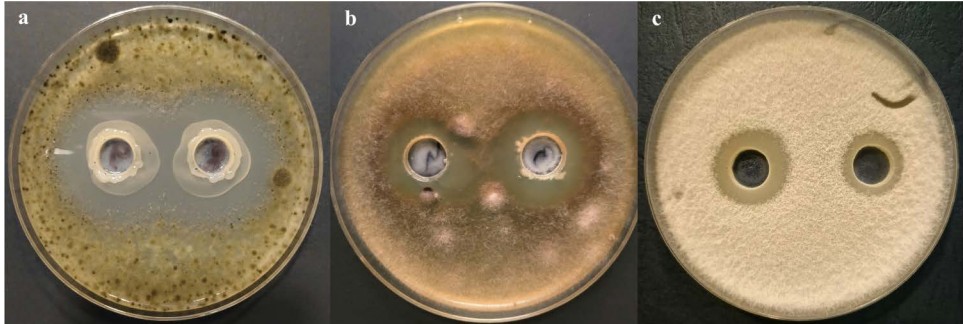

**Fig 1. Antagonistic activity of *B. velezensis* KT27 against (a) *C. beticola*, (b) *R. cerealis*, and (c) *F. oxysporum*.** In each image, the left well represents the bacterial cell suspension, while the right well contains the sterile culture supernatant.

## Quantitative assessment of phosphate solubilization

The ability of *B. velezensis* KT27 to solubilize inorganic phosphate was assessed quantitatively. After 48 hours of cultivation, bacterial cells were harvested by centrifugation at 4,500 rpm for 10 minutes to eliminate residual soluble phosphate from the culture medium. The supernatant was discarded, and the bacterial biomass was resuspended in 6 mL of sterile distilled water. The inoculum (2 mL) was then introduced into 100 mL of Pikovskaya's (PVK) liquid medium containing 2.5 g/L of tricalcium phosphate (TCP), serving as an insoluble phosphorus source [39]. Cultures were incubated at 30°C for 14 days under shaking conditions. At regular intervals, samples were aseptically collected and centrifuged (15 minutes at 4,500 rpm) to remove insoluble material. Soluble phosphate in the supernatant was quantified using the SPECTRO-QUANT Phosphate Cell Test (cat. no.: 1.00673.0001; detection range: 9–307 mg/L $PO_4^{3-}$; Merck, Darmstadt, Germany). For each measurement, 0.2 mL of sample was added to the test vial, followed by one dose of the P-1K reagent. The vial was heated at 120°C for 30 min in a Spectroquant TR 420 thermoreactor. After cooling, 5 drops of P-2K and one dose of P-3K reagent were sequentially added, mixed, and left to stand for 5 minutes. Absorbance was measured using a Spectroquant Pharo 100 spectrophotometer. Results were expressed as mg/L of soluble phosphate [40,41].

## Quantitative assessment of potassium solubilization

Potassium-solubilizing potential of *B. velezensis* KT27 was evaluated using the same inoculum preparation as for phosphate solubilization. The strain was inoculated into Aleksandrov's broth medium containing 2 g/L of muscovite [$KAl_2(AlSi_3O_{10})(OH)_2$], serving as an insoluble potassium source [42]. Cultures were incubated at 30°C for 14 days under shaking conditions. At designated time points, culture samples were collected, centrifuged, and the supernatants were analyzed using the SPECTROQUANT Potassium Cell Test (cat. no.: 1.14562.0001; detection range: 5.0–50.0 mg/L K; Merck, Darmstadt, Germany). This turbidimetric assay is based on the photometric detection of a precipitate formed when potassium ions react with sodium tetraphenylborate in an alkaline solution. For each analysis, 2.0 mL of the supernatant was transferred into the test vial, followed by the addition of 6 drops each of reagents K-1K and K-2K. The mixture was gently inverted for homogenization and incubated at room temperature for 5 minutes. Absorbance was measured using the Spectroquant Pharo 100 spectrophotometer with automatic barcode recognition. Results were expressed in mg/L of solubilized potassium.

## Quantitative assessment of zinc solubilization

Zinc solubilization by *B. velezensis* KT27 was assessed using the SPECTROQUANT Zinc Cell Test (cat. no.: 1.14566.0001; range: 0.20–5.00 mg/L Zn; Merck, Darmstadt, Germany). The assay is based on the colorimetric reaction of

zinc ions with pyridylazo-resorcinol (PAR) under alkaline conditions, producing a measurable color change. The bacterium was cultivated in Bunt and Rovira broth medium supplemented with 1 g/L of zinc oxide (ZnO), as described by previous studies [43]. Cultures were incubated at 30°C for 14 days with periodic sampling. Supernatants, obtained after centrifugation, were used for zinc quantification following the manufacturer's instructions. Absorbance was measured using the Spectroquant Pharo 100 spectrophotometer, and zinc concentrations were reported in mg/L.

### Production of *B. velezensis* KT27 as a bioactive agent against *C. beticola*

The large-scale cultivation of *B. velezensis* KT27 was performed in 5-liter stirred-tank bioreactors (Biostat B Plus, Sartorius Stedim Biotech GmbH, Germany), using a modified Tryptone Soya Broth (TSB) medium supplemented with 20 g/L of mannitol and 5 g/L of yeast extract to enhance biomass and metabolite production. A working volume of 2 L was sterilized at 121°C for 30 minutes in an autoclave. After installation and equilibration at the cultivation temperature (30°C), the medium was inoculated with *B. velezensis* KT27 at 5% (v/v), corresponding to an initial cell density of $1.0 \times 10^8$ CFU/mL. The fermentation was conducted at 30°C for 72 hours. The pH was maintained at 7.0 via automated titration using 0.1% NaOH or 0.1% HCl. Dissolved oxygen was regulated at 30% saturation through continuous aeration at 1 vvm and variable stirring between 100 and 500 rpm, as described by Rahbani Mounsef et al. [44].

To enhance the biosynthesis of antifungal metabolites, parallel fermentations were conducted under identical conditions but with the addition of 200 mL of heat-inactivated (121°C for 30 minutes) fungal culture (*C. beticola*, *F. oxysporum*, or *R. cerealis*) immediately following bacterial inoculation. Samples were periodically collected to monitor antifungal activity and to quantify the production of bioactive metabolites. For the preparation of field-grade biocontrol formulations, *B. velezensis* KT27 was cultivated under analogous conditions in a 30-liter bioreactor (Biostat C Plus, Sartorius Stedim Biotech GmbH, Germany), ensuring scalability and reproducibility of the biopreparation process [45].

### Quantitative determination of surfactin, iturin, and fengycin

The production of lipopeptide biosurfactants surfactin, iturin, and fengycin by *B. velezensis* KT27 was quantified using high-performance liquid chromatography (HPLC). Cultures (50 mL) harvested after 48 hours were centrifuged at 4,600 rpm for 20 minutes. The supernatant was collected and adjusted to 25% (v/v) methanol (Merck, Darmstadt, Germany), then purified using Solid Phase Extraction (SPE) columns (Agilent Technologies, Inc., Santa Clara, CA, USA). Following elution, the solvent was evaporated under reduced pressure and the residue was reconstituted in 1 mL of methanol. Prior to chromatographic analysis, samples were filtered through 0.22 µm nylon syringe filters (Merck, Darmstadt, Germany). HPLC analysis was performed using an Agilent 1200 Series A system (Agilent Technologies, Santa Clara, CA, USA) equipped with a Lichrosphere 100 RP18 column (250 mm × 4 mm; Merck, Darmstadt, Germany). The mobile phase consisted of solvent A: 0.76 mM trifluoroacetic acid (TFA) in water, and solvent B: 0.76 mM TFA in acetonitrile. The elution gradient was as follows: 0 min – 45% B, 3 min – 50% B, 8 min – 80% B, 25 min – 100% B, 30 min – 100% B, 31 min – 45% B. The flow rate was maintained at 1 mL/min, and the column temperature was set to 30 °C. Detection was carried out at 210 nm. Quantification was achieved using external standards, and data analysis was performed using ChemStation for LC 3D Systems software (Agilent Technologies, Inc., Santa Clara, CA, USA) [46].

### Field evaluation of *B. velezensis* KT27 based biocontrol agent for managing CLS in sugar beet

A field experiment was conducted in Winna Góra, Wielkopolskie Province, Poland (GPS: 52°12'41.2"N, 17°26'09.5"E), to evaluate the efficacy of a biological control agent based on *B. velezensis* KT27 against Cercospora leaf spot (CLS) in sugar beet. Meteorological conditions during the growing season are summarized in Table 1.

The soil at the experimental site was classified as quality class IIIb, with a pH of 6.0 and organic matter content of 1.7%. The study employed a randomized block design with four replicates per treatment. Each plot measured 3 m × 10 m (30 m²). Standard agronomic practices for sugar beet cultivation were followed. The sugar beet cultivar 'Kujavia' was sown

**Table 1. Meteorological conditions during the research.**

| Month-year | Weather parameters | Decade | | | Average/total |
|---|---|---|---|---|---|
| | | I | II | III | |
| 03-2023 | Average temperature [°C] | 0.74 | 5.88 | 8.08 | **4.90** |
| | Average air humidity [%] | 91.16 | 77.83 | 79.60 | **82.86** |
| | Total rainfall [mm] | 8.40 | 0.50 | 0.00 | **8.90** |
| 04-2023 | Average temperature [°C] | 4.85 | 9.50 | 10.42 | **8.21** |
| | Average air humidity [%] | 81.35 | 85.55 | 72.83 | **79.83** |
| | Total rainfall [mm] | 0.30 | 0.00 | 0.00 | **0.30** |
| 05-2023 | Average temperature [°C] | 10.20 | 13.71 | 16.13 | **13.35** |
| | Average air humidity [%] | 66.12 | 75.35 | 63.11 | **68.19** |
| | Total rainfall [mm] | 12.00 | 16.30 | 0.10 | **28.40** |
| 06-2023 | Average temperature [°C] | 18.40 | 18.71 | 20.08 | **19.06** |
| | Average air humidity [%] | 53.47 | 66.15 | 74.79 | **64.80** |
| | Total rainfall [mm] | 0.20 | 6.60 | 36.30 | **43.10** |
| 07-2023 | Average temperature [°C] | 20.66 | 21.65 | 18.83 | **20.38** |
| | Average air humidity [%] | 61.06 | 62.34 | 77.30 | **66.90** |
| | Total rainfall [mm] | 1.70 | 10.20 | 26.90 | **38.80** |
| 08-2023 | Average temperature [°C] | 17.51 | 23.58 | 19.28 | **20.12** |
| | Average air humidity [%] | 78.89 | 69.78 | 81.38 | **76.68** |
| | Total rainfall [mm] | 36.50 | 2.70 | 25.80 | **65.00** |
| 09-2023 | Average temperature [°C] | 19.10 | 19.24 | 17.65 | **18.66** |
| | Average air humidity [%] | 72.96 | 73.11 | 75.40 | **73.82** |
| | Total rainfall [mm] | 0.00 | 6.20 | 5.70 | **11.90** |
| 10-2023 | Average temperature [°C] | 13.01 | 9.92 | 11.05 | **11.33** |
| | Average air humidity [%] | 66.10 | 83.06 | 92.83 | **80.67** |
| | Total rainfall [mm] | 11.00 | 8.10 | 17.50 | **36.60** |

on April 6, 2023, with row spacing of 45 cm. Fertilization was tailored to site-specific soil fertility and crop nutrient requirements, as outlined in Table 2. The trial was conducted following European and Mediterranean Plant Protection Organization (EPPO) guidelines: PP 1/1(4), PP 1/135(4), PP 1/152(4), PP 1/181(5), and PP 1/225(2) [47–50].

Control measures for CLS were applied three times throughout the growing season. Chemical control treatments included the fungicides Dafne 250 EC (difenoconazole, 250 g/L; INNVIGO Sp. z o.o., Poland), Makler 250 SE (azoxystrobin, 250 g/L; INNVIGO Sp. z o.o., Poland), and Tebu 250 EW (tebuconazole, 250 g/L; Helm AG, Germany). The biological treatment involved the application of *B. velezensis* KT27, either as a standard culture or as an induced culture cultivated in the presence of inactivated *C. beticola, F. oxysporum,* or *R. cerealis* to stimulate the production of antifungal metabolites. Application schedules and dosages for both chemical and biological treatments are presented in Table 3.

Treatments were applied using a calibrated backpack sprayer equipped with TeeJet 110 03 VXR nozzles. The working parameters were set to a spray volume of 200 L/ha, boom height of 50 cm above the canopy, and a travel speed of 5 km/h. Herbicide and insecticide treatments were administered according to current agronomic recommendations (Table 3).

Biological control efficacy was assessed by estimating the percentage of foliar area affected by CLS on 25 randomly selected plants per plot. Visual assessments were performed according to the EPPO standard PP 1/1(4) methodology [51]. Disease severity in treated plots was compared with untreated controls, and results were expressed as percentages of affected leaf area. Root yield was determined on October 10, 2023, by harvesting the entire plot and weighing the roots on a calibrated technical scale. Yields were converted to tons per hectare.

**Table 2. Fertilization timing and dosages in sugar beet cultivation.**

| Application dates | Nutrients concentration (kg x ha⁻¹) | | | |
|---|---|---|---|---|
| | N | P | K | S |
| 28.03.2023 | 50,6 | | | |
| | 15 | 50 | 75 | 17,5 |
| | | | 140 | |
| | | 45 | | |
| 12.05.2023 | 54,4 | | | |

**Table 3. Timing and dosages of chemical fungicides and biological treatments in sugar beet cultivation.**

| No. | Fungicide/ biocontrol agents | Application data (T1, T2, T3) | Dose of fungicide/ biocontrol agents (L x ha⁻¹) |
|---|---|---|---|
| 1 | Control (no protection) | – | – |
| 2 | T1 - difenoconazole | T1-27.07.2023 | T1 - 0.4 |
| | T2 - azoxystrobin | T2-17.08.2023 | T2 - 1.0 |
| | T3 – tebuconazole | T3-02.09.2023 | T3 - 0.8 |
| 3 | T1, T2, T3 – *B. velezensis* KT27 | T1-27.07.23 T2-10.08.23 T3-25.08.23 | T1, T2, T3 - 0.5 |
| 4 | T1, T2, T3 - *B. velezensis* KT27 + *C. beticola* | | T1, T2, T3 - 0.5 |
| 5 | T1, T2, T3 - *B. velezensis* KT27 + *R. cerealis* | | T1, T2, T3 - 0.5 |
| 6 | T1, T2, T3 - *B. velezensis* KT27 + *F. oxysporum* | | T1, T2, T3 - 0.5 |

Data were statistically analyzed using one-way analysis of variance (ANOVA) to evaluate treatment effects. Post-hoc pairwise comparisons were conducted using Tukey's Honest Significant Difference (HSD) test at a significance level of $p \le 0.05$. Statistical analyses were performed using OriginPro 2024b (OriginLab Corporation, USA). Results are presented as mean ± standard deviation, with significant differences between groups indicated in figures or tables. The efficacy of each treatment was calculated using Abbott's formula.

## Results

### Morphological characterization and identification of the isolated bacterial strain

The morphological characteristics of the bacterial strain KT27 were assessed through colony morphology and microscopic examination. The strain was cultured on tryptic soy agar (TSA) and incubated at 30°C for 48 hours. Colonies appeared circular, opaque, and cream to white in color, with rough surfaces and slightly undulate or irregular margins. Their texture ranged from dry to slightly mucoid. Gram staining and light microscopy revealed Gram-positive rod-shaped cells, occurring singly or in short chains, with centrally or subterminally located endospores. Preliminary identification using matrix-assisted laser desorption/ionization time-of-flight mass spectrometry (MALDI-TOF MS) suggested the strain belonged to the genus *Bacillus*. However, the obtained identification score (1.81) was below the threshold for reliable species level classification. For definitive identification, 16S rRNA gene sequencing was performed. Sequence analysis revealed 100% identity with reference sequences for *B. velezensis*, confirming the taxonomic status of the isolate. The strain, designated *B. velezensis* KT27, has been deposited in the Polish Collection of Microorganisms at the Institute of Immunology and Experimental Therapy in Wrocław under accession number B/00463. The collection is recognized by both the World Federation for Culture Collections (WFCC No. 106) and the European Culture Collections' Organisation (ECCO).

## Antifungal activity of *B. velezensis* KT27

Laboratory assays demonstrated that the bacterial strain *B. velezensis* KT27 exhibits pronounced antagonistic activity against *C. beticola*, the phytopathogen responsible for Cercospora leaf spot (CLS), a serious disease affecting sugar beet (*B. vulgaris*) and other members of the *Asteraceae* family. These findings underscore the remarkable biocontrol potential of *Bacillus* spp., supporting their application as biological components in integrated plant protection strategies [52]. Using the agar well diffusion method to assess antifungal activity, a substantial inhibition zone was observed surrounding wells inoculated with *B. velezensis* KT27, indicating strong suppression of *C. beticola* growth (Fig 2). The most potent inhibitory effect—measured as 60.2% reduction in fungal growth—was achieved with the whole cell culture of *B. velezensis* KT27 incubated for 48 hours. In contrast, when bacterial cells were removed via centrifugation and sterile filtration (ø 0.22 µm), the inhibitory effect decreased by approximately 69.2%, suggesting that both bacterial cells and their metabolites contribute synergistically to antifungal activity.

In comparison, the inhibitory effects of *B. velezensis* KT27 on the growth of other phytopathogenic fungi *F. oxysporum* and *R. cerealis* were considerably weaker. Specifically, inhibition against *F. oxysporum* and *R. cerealis* was 3.9- and 2.7-fold lower, respectively, than that observed for *C. beticola*. Furthermore, culture supernatants alone exhibited markedly reduced activity, with more than a 3.5-fold decrease in growth inhibition, highlighting the limited efficacy of extracellular metabolites in isolation.

Additional experiments were conducted to evaluate whether exposure to inactivated fungal cells could enhance the antagonistic potential of *B. velezensis* KT27. Heat-inactivated cultures of *C. beticola*, *F. oxysporum*, and *R. cerealis* were introduced into *B. velezensis* KT27 cultures to investigate possible priming or induction effects. The results revealed that all tested inactivated fungi, with the exception of *F. oxysporum*, stimulated an increase in the antifungal activity of *B. velezensis* KT27 (Fig 3). Notably, the greatest enhancement of antagonistic activity against *C. beticola* was elicited not by its own inactivated cells, but by inactivated *R. cerealis*, which induced a 33.5% increase in inhibition. In contrast, the addition of inactivated *C. beticola* resulted in a comparatively modest 13.7% increase over the uninduced control culture.

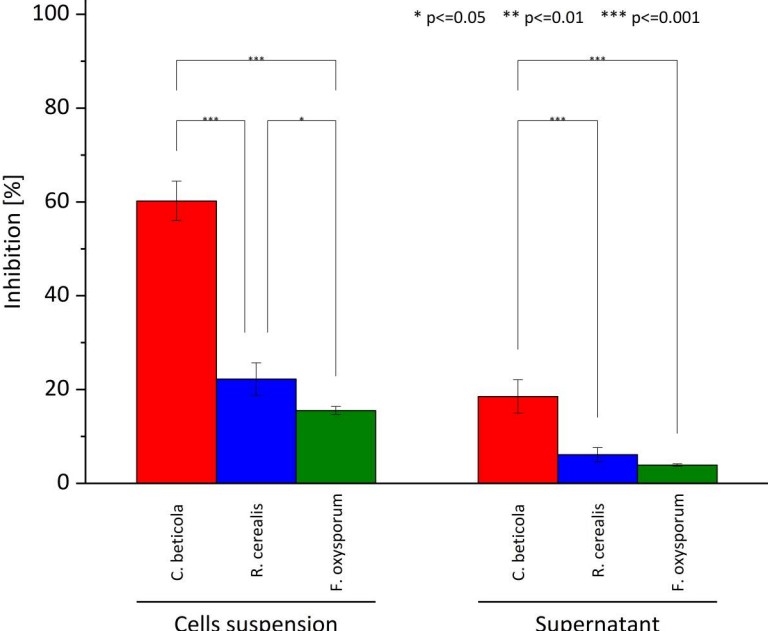

**Fig 2. Antifungal activity of *B. velezensis* KT27 against *C. beticola*, *R. cerealis*, and *F. oxysporum*.** The graph illustrates the inhibitory effects both cell suspensions and cell-free supernatant fractions. Statistical significance is denoted as follows: $p \leq 0.05$ (*), $p \leq 0.01$ (**), and $p \leq 0.001$ (***).

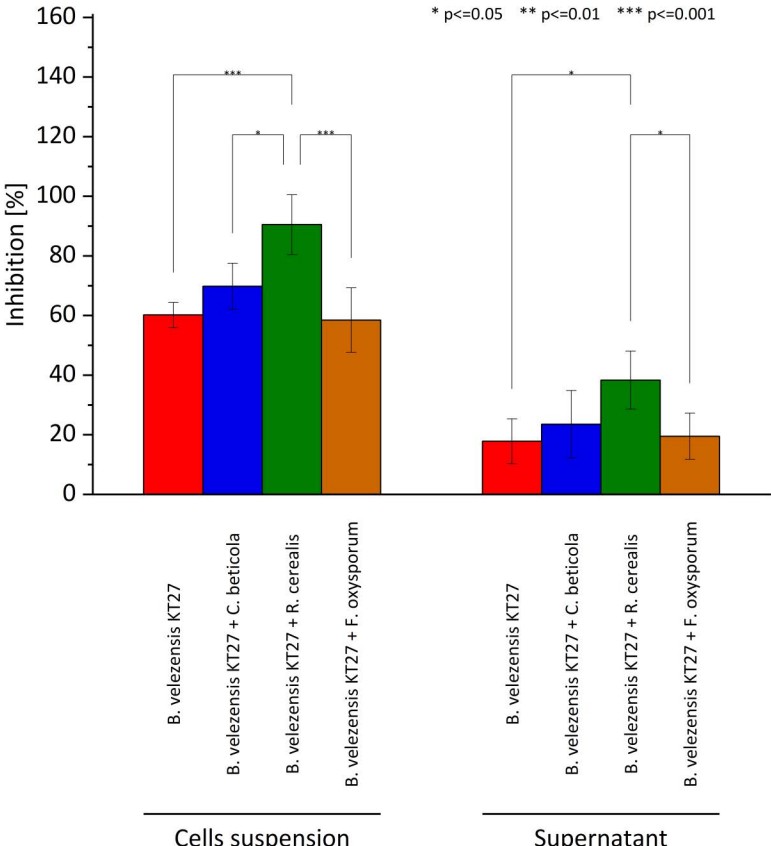

**Fig 3. Antifungal activity of *B. velezensis* KT27 against *C. beticola* with and without induction by inactivated fungal cultures.**

A similar pattern was observed when testing cell-free supernatants from these induced cultures. The supernatant from *B. velezensis* KT27 cultured with inactivated *R. cerealis* exhibited the highest enhancement in antifungal activity, further supporting the hypothesis that induction by non-host fungal signals can potentiate the antifungal arsenal of *B. velezensis* KT27.

## Solubilization of phosphorus, potassium, and zinc by *B. velezensis* KT27

Investigations into the mineral solubilizing capacity of *B. velezensis* KT27 revealed a substantial increase in the availability of soluble forms of phosphorus (P), potassium (K), and zinc (Zn), all of which are essential for optimal plant growth and yield. The bacterium was cultured for 14 days on selective media supplemented with insoluble mineral sources, allowing assessment of its ability to mobilize these nutrients into plant available forms (Fig 4). The data confirmed that *B. velezensis* KT27 possesses significant solubilization capabilities, particularly for phosphorus and potassium. The release of these elements closely paralleled bacterial growth, as measured by optical density at 600 nm ($OD_{600}$). In the early stages of cultivation, concentrations of soluble phosphorus and potassium ranged from 5.4 to 10.4 mg/L and 5.2 to 15.7 mg/L, respectively. As the culture progressed into the logarithmic growth phase, the concentrations of both elements increased rapidly. Peak solubilization was recorded at 36 hours, reaching 53.3 mg/L for phosphorus and 45.8 mg/L for potassium, with potassium solubilization occurring more rapidly than phosphorus. Following the transition into the stationary phase, where $OD_{600}$ values plateaued, no further increase in phosphorus or potassium solubilization was observed, indicating a growth-dependent mechanism for mineral release.

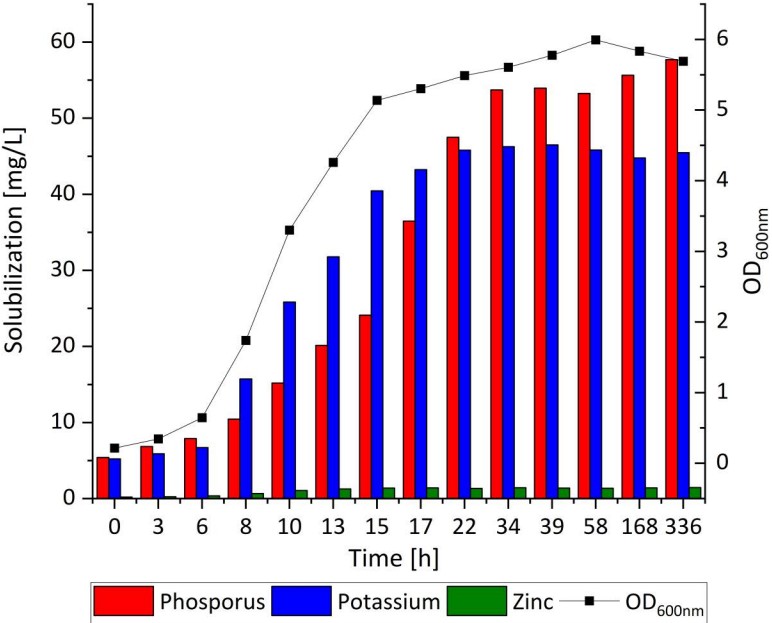

**Fig 4. Solubilization of phosphorus, potassium, and zinc by *B. velezensis* KT27 in relation to bacterial growth kinetics.**

In contrast, zinc solubilization and release by *B. velezensis* KT27 was markedly lower. Across the cultivation period, the concentration of soluble zinc ranged from 0.2 to 6.0 mg/L. This suggests either a limited enzymatic or metabolic capacity of the strain for zinc mobilization, or a slower mechanism compared to that of phosphorus and potassium.

## Quantification of surfactin, iturin, and fengycin in *B. velezensis* KT27 cultures

Chromatographic analysis of culture supernatants from *B. velezensis* KT27 both under standard conditions and in the presence of heat inactivated fungal pathogens (*C. beticola*, *F. oxysporum*, and *R. cerealis*) confirmed the production of three key lipopeptides: surfactin, iturin, and fengycin. These secondary metabolites are well documented for their roles in antimicrobial activity and plant protection. Overall, the total concentrations of the detected lipopeptides were relatively low (Fig 5). Among them, surfactin was the most abundant, with concentrations ranging from 10.2 to 12.8 mg/L depending on the specific culture conditions. In contrast, significantly lower levels of iturin and fengycin were observed, with maximum concentrations not exceeding 2.0 mg/L. These findings indicate that *B. velezensis* KT27 produces detectable but modest amounts of antifungal lipopeptides under the tested conditions, with surfactin as the dominant compound. Further optimization of culture conditions or the use of stronger fungal inducers may enhance lipopeptide synthesis and improve biocontrol efficacy.

## Control of CLS in sugar beet: Field evaluation of *B. velezensis* KT27 based biocontrol

An essential component of this research was the evaluation of a biocontrol formulation containing *B. velezensis* KT27 for the suppression of CLS under field conditions. To this end, 72-hour bioreactor cultures of the bacterial strain were prepared, including non-induced cultures and those induced with heat inactivated fungal pathogens (*C. beticola*, *F. oxysporum*, and *R. cerealis*). These bioactive formulations were applied as foliar treatments in sugar beet crops. The culture medium was optimized by supplementing TSB with 20 g/L mannitol and 5 g/L yeast extract, resulting in a high-density bacterial suspension ($1.0 \times 10^9$ CFU/mL). The treatments were administered directly to infected plants

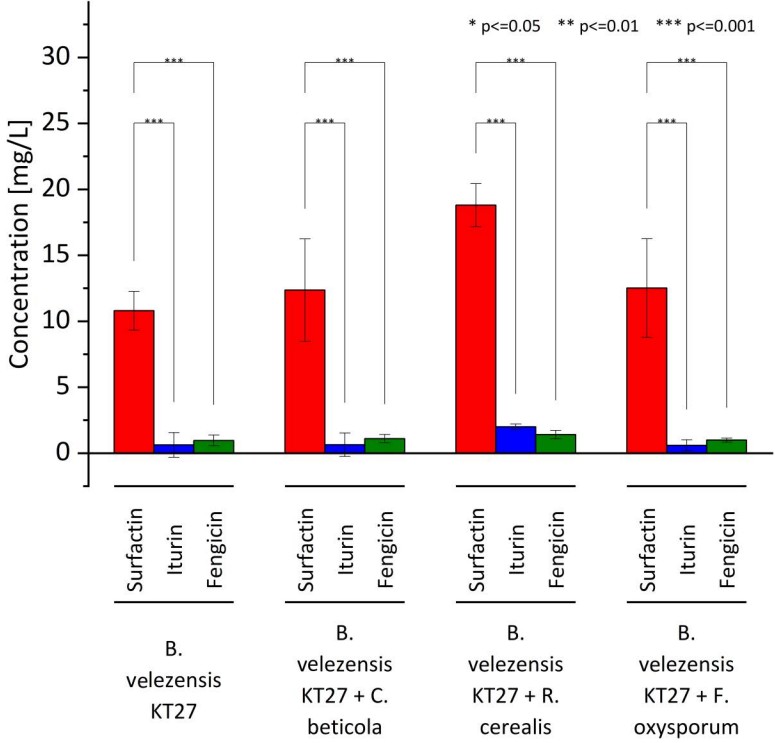

**Fig 5. Quantification of key lipopeptides (surfactin, iturin, and fengycin) in cultures of *B. velezensis* KT27 with and without induction by heat-inactivated *C. beticola*, *R. cerealis*, and *F. oxysporum*.**

and compared with untreated controls and chemical fungicides (difenoconazole, azoxystrobin, tebuconazole). Field trials demonstrated that *B. velezensis* KT27 significantly reduced CLS symptoms and inhibited the spread of *C. beticola* (Fig 6). As observed in laboratory assays, the most effective field treatment involved the use of *B. velezensis* KT27 cultures induced with inactivated *R. cerealis*, achieving a CLS suppression rate of up to 96.4% at the first application (T1) compared to untreated control. Notably, this biological treatment outperformed chemical control (difenoconazole) by approximately 11% at T1. Although the efficacy declined slightly over subsequent applications (T2 and T3), substantial control was maintained, with suppression levels of 52.9% and 51.5%, respectively. Overall, the biocontrol efficacy across all terms was only 9.1% lower than that of sequential chemical treatments using three different active substances. In contrast, significantly lower control of CLS was observed in treatments utilizing *B. velezensis* KT27 cultures induced with inactivated *C. beticola* or *F. oxysporum*. The non-induced *B. velezensis* KT27 treatment resulted in a mere 12.1% reduction in CLS incidence by the final application (T3), which was over 50% less effective than the chemical standard. A consistent trend across all variants showed the highest biocontrol efficacy when treatments were applied to young plants (T1), ranging from 26.4% (non-induced *B. velezensis* KT27) to 96.4% (*B. velezensis* KT27 + *R. cerealis*) (Fig 7).

Importantly, the enhanced disease control translated into increased root yield. Regardless of the treatment variant, all biocontrol agents led to a significantly higher sugar beet yield compared to the untreated control. Despite differences in disease suppression, root yields ranged from 67.3 to 73.0 t/ha, representing a 6.5–15.2% increase over the control. The highest yield (73.0 t/ha) was recorded in plots treated with the *B. velezensis* KT27 + *R. cerealis* formulation, which was only 10.8% lower than that achieved through chemical protection (81.7 t/ha).

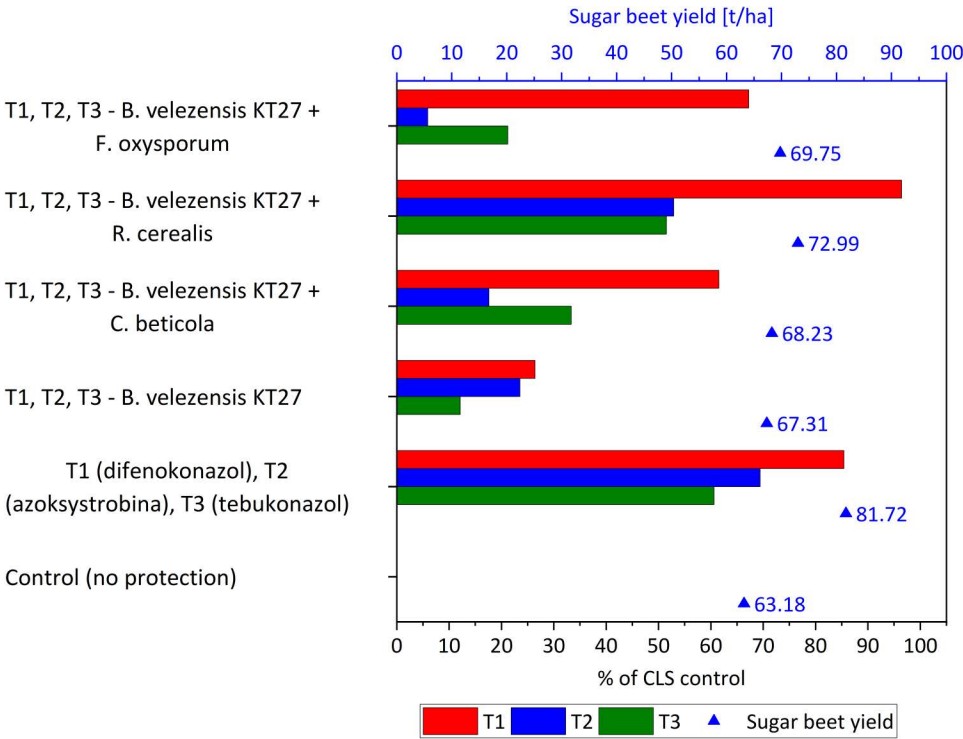

**Fig 6. Efficacy of different variants *B. velezensis* KT27 treatment in controlling CLS compared to chemical fungicides across three application terms (T1–T3), along with their impact on sugar beet root yield.**

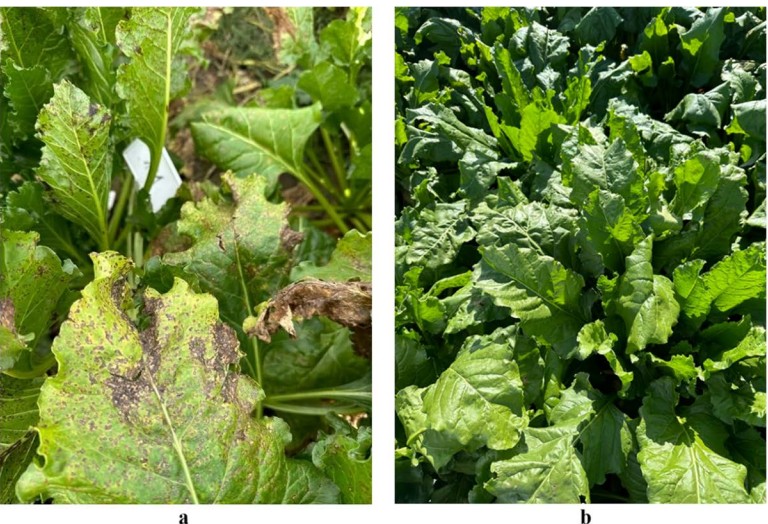

**Fig 7. Visual assessment of leaf infection: (a) untreated control, (b) plants treated with *B. velezensis* KT27 induced by inactivated *R. cerealis*.**

## Discussion

The isolation of microorganisms from compost led to the identification of a bacterial strain classified as *B. velezensis* KT27, based on morphological, metabolic, and molecular analyses. Strains of *B. velezensis* are frequently isolated from

diverse habitats and, like other members of the *Bacillus* genus, are widely recognized for their value in agricultural bio-technology and biopreparation industries [53] This species has been recovered from environments including soil, compost, fermented foods, and various plant tissues. Notably, the rhizosphere is a key reservoir, where *B. velezensis* strains contribute to plant growth and disease suppression through the synthesis of phytohormones and antimicrobial compounds [54]. Compost, with its nutrient-rich and competitive conditions, promotes the proliferation of metabolically active, spore forming bacteria such as *B. velezensis* [20]. Additionally, its presence in fermented foods (e.g., *natto*) underlines its safety for human use, while its endophytic forms offer benefits in plant stress resistance and nutrient mobilization [53].

Effective control of phytopathogenic fungi is vital in modern agriculture, as fungal infections drastically reduce crop productivity, compromise harvest quality, and, in severe cases, render produce unmarketable. The economic burden of fungal diseases is substantial on both global and local scales, impacting food security and rural livelihoods. Cercospora leaf spot (CLS), caused by *C. beticola*, is one such disease with considerable agronomic significance. Among the most promising disease management strategies is the use of biological control agents—particularly members of the *Bacillus* genus—which exhibit broad spectrum antagonism against phytopathogenic fungi [14,15,17,18,55]. In this study, we evaluated the potential of *B. velezensis* KT27 as a sustainable and ecological approach for controlling *C. beticola* in sugar beet cultivation. In addition to disease suppression, we also investigated traits associated with plant growth promotion. Laboratory assays demonstrated the strain's capacity to solubilize key nutrients (phosphorus, potassium, and zinc) and inhibit *C. beticola* growth in vitro. These findings were further validated under field conditions, confirming the dual functionality of the strain. Prior research has shown that the antagonistic activity of beneficial microbes is influenced by environmental and biological factors [5,10]. Our results revealed that the antifungal potential of *B. velezensis* KT27 could be significantly enhanced through the addition of thermally inactivated fungal biomass, particularly from *R. cerealis*. This induction strategy led to more than a 50% increase in antifungal activity, likely due to the stimulation of bacterial defense mechanisms. Similar findings have been reported in the literature. For example, de Boer et al. demonstrated that *Mucor hiemalis* enhanced the antagonistic potential of *Achromobacter* and *Stenotrophomonas* against *R. solani* [56]. The underlying mechanism may involve chitinase-mediated degradation of fungal cell walls, a hypothesis also supported by studies involving *Plasmodiophora brassicae* spore induction [57,58].

Metabolomic profiling of *B. velezensis* KT27 cultures confirmed the production of cyclic lipopeptides (CLPs), notably surfactin, fengycin, and iturin compounds are well documented for their roles in microbial antagonism [59–61]. These lipopeptides disrupt fungal membranes by increasing permeability, impairing membrane integrity, and ultimately inhibiting spore germination or causing hyphal lysis [62]. Among the detected compounds, surfactin was consistently produced at the highest levels, although the concentrations (18.8 mg/L) were modest compared to literature reports of several grams per liter [63]. While surfactin is less commonly associated with direct antifungal activity compared to iturin or fengycin, its role as a facilitator of membrane disruption and inducer of host defense responses is increasingly recognized. For instance, surfactin from *B. subtilis* SF1 inhibited *F. foetens* mycelial growth by over 50%, both in vitro and in vivo [64]. The authors also noted increased antioxidant activity and altered protein expression profiles. Remarkably, the antifungal activity of this bacteria remained largely unaffected after heating to 100°C, suggesting the presence of thermostable bioactive metabolites, possibly beyond enzymatic degradation pathways. Surfactin may also enhance the penetration and activity of co-produced lipopeptides like fengycin and iturin, despite not directly targeting fungal DNA [65]. Although in vitro efficacy is a valuable first step, field validation is critical for real world application. Our *in vivo* experiments demonstrated that foliar application of *B. velezensis* KT27 suspensions significantly reduced CLS severity in sugar beet. The most effective treatment *B. velezensis* KT27 induced by *R. cerealis*—achieved approximately 50% disease suppression, with efficacy only 9.1% lower than that of combined chemical fungicides (difenoconazole, azoxystrobin, tebuconazole). Moreover, this biological treatment resulted in a root yield increase of up to 15% relative to untreated controls, likely due to both disease mitigation and improved nutrient bioavailability via phosphorus and potassium solubilization. These findings align with prior studies highlighting the biocontrol potential of *Bacillus* spp. El Housni et al. screened 18 microbial isolates for

antagonism against *C. beticola*, identifying that 50% exhibited antifungal activity, with some inhibiting growth by up to 80% [66]. Among the active isolates, four belonged to the *Bacillus* genus and were confirmed to harbor genes involved in CLP biosynthesis. In their field trials, *Bacillus* based formulations not only suppressed CLS by 77.4% but also promoted plant growth—results comparable to those obtained with chemical fungicides (88.5%) [67]. Similarly, Arzanlou et al. reported 60.9% CLS control by *Bacillus* strain RB2 in field trials, while greenhouse experiments showed up to 96% protection [68]. Collins et al. highlighted that vegetative cells, rather than spores, were more effective in inducing sugar beet resistance to *C. beticola* [19]. Esh et al. further demonstrated that *B. amyloliquefaciens* strains isolated from the phyllosphere were most effective when applied preventively [68]. Soliman et al. also showed ~40% efficacy of *B. amyloliquefaciens* RaSh1 against *Alternaria alternata*, another leaf pathogen [69]. Conversely, some studies report limited protection using *B. subtilis* alone, suggesting that efficacy is strain-specific and influenced by formulation and environmental compatibility [70].

## Conclusion

Effective protection of cultivated plants against fungal pathogens is essential for maintaining stable crop yields and ensuring the economic viability of agricultural systems. However, the extensive use of chemical plant protection agents presents considerable risks to environmental health, biodiversity, and, in many cases, to the safety and quality of food products. These concerns have fueled growing interest in the development of biologically based alternatives capable of mitigating both the incidence and consequences of plant diseases.

Among the most promising candidates for biological control are bacteria of the genus *Bacillus*, with *B. velezensis* demonstrating particularly strong potential. These microorganisms are readily cultivable, exhibit high resilience to abiotic stressors, and are capable of forming endospores, which enhances their environmental persistence and stability. In our study, *in vitro* assays followed by field validation confirmed that *B. velezensis* KT27 effectively inhibits the growth of *C. beticola*, the causal agent of Cercospora leaf spot (CLS) in sugar beet. Moreover, the application of this strain not only reduced disease severity but also improved crop yield, likely due to its ability to mobilize plant available nutrients underscoring its potential as a biofertilizer component.

Our findings also emphasize the complex interplay of microbial factors governing antifungal efficacy. We demonstrated that culturing *B. velezensis* KT27 in the presence of inactivated *R. cerealis* biomass significantly enhances lipopeptide production and antifungal activity. This suggests that fungal derived elicitors may stimulate secondary metabolite biosynthesis in bacteria, thereby augmenting its biocontrol properties. Such induced formulations may represent a novel strategy for optimizing the efficacy of biological control agents.

Taken together, these results position *B. velezensis* KT27 as a multifunctional agent capable of both pathogen suppression and plant growth promotion. The integration of such biologically derived products into conventional agricultural practices could play a pivotal role in advancing sustainable crop management, reducing reliance on synthetic fungicides and mineral fertilizers, and promoting a more environmentally responsible approach to plant protection.

## Author contributions

**Conceptualization:** Roman Marecik, Wojciech Białas, Katarzyna Czaczyk, Agnieszka Drożdżyńska, Łukasz Sobiech.

**Funding acquisition:** Łukasz Sobiech, Ewa Jajor.

**Investigation:** Roman Marecik, Agnieszka Wita, Wojciech Białas, Agnieszka Drożdżyńska, Łukasz Sobiech, Monika Grzanka, Jakub Danielewicz, Ewa Jajor, Joanna Horoszkiewicz.

**Methodology:** Jakub Danielewicz, Ewa Jajor, Joanna Horoszkiewicz.

**Supervision:** Wojciech Białas, Łukasz Sobiech.

**Writing – original draft:** Roman Marecik, Wojciech Białas, Łukasz Sobiech.

**Writing – review & editing:** Monika Grzanka.

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
