## [Decision Letter · Decision Letter 0]

26 Feb 2025

PONE-D-25-07815Use of Bacillus valezensis KT 27 induced by fungi in the protection of sugar beets against Cercospora Leaf Spot (CLS)PLOS ONE

Dear Dr. Marecik,

Thank you for submitting your manuscript to PLOS ONE. After careful consideration, we feel that it has merit but does not fully meet PLOS ONE’s publication criteria as it currently stands. Therefore, we invite you to submit a revised version of the manuscript that addresses the points raised during the review process.

We look forward to receiving your revised manuscript.

Kind regards,

Estibaliz Sansinenea

Academic Editor

PLOS ONE

Journal Requirements:

2.  We note that your Data Availability Statement is currently as follows: 

“All relevant data are within the manuscript and its Supporting Information files.”

**Additional Editor Comments:**

The reviewers have commented about the Ms and reccommended major revision before its acceptation. I invite the authors to respond carefully all reviewers comments doing the neccessary changes .

Reviewers' comments:

Reviewer's Responses to Questions

**Comments to the Author**

1. Is the manuscript technically sound, and do the data support the conclusions?

Reviewer #1: Partly

Reviewer #2: Yes

2. Has the statistical analysis been performed appropriately and rigorously? 

Reviewer #1: No

Reviewer #2: Yes

3. Have the authors made all data underlying the findings in their manuscript fully available?

Reviewer #1: No

Reviewer #2: Yes

4. Is the manuscript presented in an intelligible fashion and written in standard English?

Reviewer #1: No

Reviewer #2: No

5. Review Comments to the Author

Reviewer #1: 1-The submitted manuscripts has significant technical weaknesses and lacks novelty. The writing quality is poor, and the ideas are repetitive. The researchers used a single bacterial strain alongside three pesticides, which creates an unfair comparison. It would have been more appropriate to include multiple bacterial strains for a more balanced evaluation. Additionally, the methodology requires substantial revision to ensure clarity and robustness. Must of the results i can't believe it .2-The images in Figure 1 indicate the absence of a proper control treatment. Although the researcher claims that one of the wells contains a control treatment, a thorough review reveals that the so-called control treatment actually impacts the fungus. This is both unacceptable and ethically inappropriate.3-Please revise the research and simplify the sentences for clarity. The section on materials and methods needs to be rewritten in a clear and organized manner. The results are not presented in a logical sequence and should be restructured accordingly. Additionally, the discussion is weak and requires improvement. It is recommended to incorporate the suggested sources identified in the research to strengthen the discussion.4-The discussion section requires the addition of relevant sources and the removal of those that are not directly applicable to the topic.5- No images of molecular analysis, field experiment, images of treated and untreated plants, results of MALDI-TOF MS .

Reviewer #2: There are some suggestions and recommendations to improve the study impact.

• Further Studies on Mechanisms: Future research should focus on elucidating the precise mechanisms of action of Bacillus velezensis KT27, including the roles of specific metabolites in antagonism.

• Broader Field Trials: Expanding the field trials to different geographical locations and crop types could help validate the versatility of the strain as a biocontrol agent under diverse environmental conditions.

• Integrated Pest Management (IPM): Research should investigate the combination of Bacillus velezensis KT27 with other biocontrol agents or cultural practices in Integrated Pest Management (IPM) systems to assess its compatibility and potential synergistic effects.

• Ecotoxicological Assessments: More comprehensive studies on the environmental impact of the strain, including its effects on non-target organisms and soil health, would be beneficial for its eventual commercial application.

6. PLOS authors have the option to publish the peer review history of their article (what does this mean? ). If published, this will include your full peer review and any attached files.

**Do you want your identity to be public for this peer review?** For information about this choice, including consent withdrawal, please see our Privacy Policy .

Reviewer #1: No

Reviewer #2: No

---

## [Author Response · Author response to Decision Letter 1]

14 Apr 2025

Dear Reviewers,

We would like to express our deepest gratitude for your thoughtful and constructive comments regarding our manuscript. Your insightful suggestions have been of great value to us, and we sincerely appreciate the time and effort you invested in reviewing our work.

We are confident that the revisions made in response to your remarks have considerably improved the scientific rigor, clarity, and overall quality of the manuscript. During the revision process, we also devoted careful attention to refining the language, as well as enhancing the technical and formal aspects of the manuscript.

To facilitate your review of the revised version, we have enabled the “Track Changes” feature to clearly indicate all substantial modifications. We have also included responses to your comments directly in the form of annotations. Nevertheless, we would kindly point out that the final, clean version of the manuscript best represents the full extent of the improvements made.

As part of the revision process, we have amended the manuscript title, which we believe now more accurately reflects the scope and essence of the study. Additionally, we have revised the titles of several subsections to improve structure and readability. Following your valuable suggestions, we have also updated the reference list and included photographs from the sugar beet field trials in which our bacterial biopreparation was applied.

We would also like to take this opportunity to clarify that the illustrative images provided reflect fungistatic activity observed under in vitro conditions. The well on the left side of the photograph demonstrates the activity of the bacterial biomass of Bacillus velezensis KT27, whereas the well on the right side corresponds to the post-culture fluid devoid of bacterial cells. We did not include images from the control treatment, as the application of sterile saline into wells on control plates did not result in any inhibition of fungal growth (i.e., no visible clearing zones were present). In our view, these would not contribute additional informative value.

We fully acknowledge that this is a broad and multifaceted topic, and that there are various possible approaches to its investigation. At the same time, we wish to emphasize that verifying laboratory results under field conditions—though often complex—is critical for confirming the efficacy of a biological preparation in practical agricultural settings. We stand by the reliability and validity of the results we have obtained.

While numerous studies are focused on identifying novel microbial strains with unique metabolic capacities for plant protection, we would like to highlight that the induction of fungistatic activity in microorganisms—such as the one we demonstrated—is still relatively underexplored. We sincerely hope that the publication of our findings will encourage further research others into this promising and important area.

We wholeheartedly agree with your recommendation to continue investigations along the lines outlined in the manuscript. Indeed, we are actively pursuing further research with the ultimate goal of formally registering our biopreparation, based on the B. velezensis KT27 strain, for protection against CLS, and eventually making it commercially available.

In our opinion, the subject of our research is well aligned with the increasingly urgent need to reduce the reliance on conventional chemical plant protection agents, as well as to mitigate the growing problem of pathogen resistance. These efforts, alongside a reduction in mineral fertilization, may substantially contribute to the improvement of environmental conditions impacted by modern agricultural practices.

Once again, we extend our sincere appreciation for your kind attention, careful review, and valuable input. We are truly grateful for the opportunity to improve our manuscript with your guidance.

With the highest respect and kind regards,

The Authors

---

## [Editor Report · Decision Letter 1]

16 Apr 2025

Biocontrol of Cercospora Leaf Spot in sugar beet by a novel Bacillus velezensis KT27 strain: Enhanced antifungal activity and growth promotion in laboratory and field conditions

PONE-D-25-07815R1

Dear Dr. Marecik,

We’re pleased to inform you that your manuscript has been judged scientifically suitable for publication and will be formally accepted for publication once it meets all outstanding technical requirements.

Kind regards,

Estibaliz Sansinenea

Academic Editor

PLOS ONE

Additional Editor Comments (optional):

The authors have followed all comments suggested by the reviewers therefore the MS has been improved and it can be accepted in the current form.
---

## [Editor Report · Acceptance letter]

PONE-D-25-07815R1

PLOS ONE

Dear Dr. Marecik,

I'm pleased to inform you that your manuscript has been deemed suitable for publication in PLOS ONE. Congratulations! Your manuscript is now being handed over to our production team.

Kind regards,

on behalf of

Dr. Estibaliz Sansinenea

Academic Editor

PLOS ONE